# Carbon Quantum Dots: The Role of Surface Functional Groups and Proposed Mechanisms for Metal Ion Sensing

Hasan Shabbir [1,*], Edit Csapó [2,3] and Marek Wojnicki [1]

[1] Faculty of Non-Ferrous Metals, AGH University of Science and Technology, Mickiewicza Ave. 30, 30-059 Krakow, Poland; marekw@agh.edu.pl
[2] MTA-SZTE Lendület "Momentum" Noble Metal Nanostructures Research Group, University of Szeged, Rerrich B. Sqr. 1, H-6720 Szeged, Hungary; juhaszne@chem.u-szeged.hu
[3] Interdisciplinary Excellence Center, Department of Physical Chemistry and Materials Science, University of Szeged, Rerrich B. Sqr. 1, H-6720 Szeged, Hungary
[*] Correspondence: shabbir@agh.edu.pl; Tel.: +48-729684598

**Abstract:** Carbon dots (CDs) are zero-dimensional nanomaterials composed of carbon and surface groups attached to their surface. CDs have a size smaller than 10 nm and have potential applications in different fields such as metal ion detection, photodegradation of pollutants, and bio-imaging, in this review, the capabilities of CDs in metal ion detection will be described. Quantum confinement is generally viewed as the key factor contributing to the uniqueness of CDs characteristics due to their small size and the lack of attention on the surface functional groups and their roles is given, however, in this review paper, the focus will be on the functional group and the composition of CDs. The surface functional groups depend on two parameters: (i) the oxidation of precursors and (ii) their composition. The mechanism of metal ion detection is still being studied and is not fully understood. This review article emphasizes the current development and progress of CDs, focusing on metal ion detection based on a new perspective.

**Keywords:** carbon dots; synthesis parameters; surface functional groups; metal ion sensing; mechanism; oxidation

## 1. Introduction

Carbon dots (CDs) are quasi-spherical materials mainly composed of carbon, oxygen, and hydrogen [1,2]. Their size is less than 10 nm, although few exceptions are larger than 60 nm, they categorized as zero-dimensional nanomaterials [3]. CDs consist of core and functional groups on their surface [4]. The most common functional groups are hydroxyl groups (–OH) [5], carboxyl groups (–COOH) [6], carbonyl groups (–CO) [7], and amino groups (–NH$_2$), as shown in Figure 1A [8]. Functional group manipulation and doping can be used to change the properties of the CDs surface and the core, respectively [9]. Figure 1B demonstrates that the functional groups and structure of CDs have an impact on optical characteristics and how photoluminescence redshift is affected by oxidation.

The exact properties and phenomena behind functional groups are still an open topic for research. CDs were accidentally discovered during the purification of arc-synthesized single-walled carbon nanotubes [10]. CDs have attracted a lot of attention since then due to their unique photoluminescent properties. CDs have also been investigated for their biocompatibility, sensing, and photodegradation properties. They are quasi-spherical and composed of a crystalline structure and amorphous surface [11,12] while sp$^2$ bonding is dominant in CDs, with some sp$^3$ clusters also present [13]. CDs primarily consist of carbon, hydrogen, and oxygen, but depending on the precursor, they can also contain other elements including nitrogen, boron, phosphorous, and sulfur. These elements also influence CDs physical and chemical properties [14,15]. CDs properties are tunable based on the precursor, time of synthesis, synthesis method, and temperature, due to which

there is a lot of scope for research for the exact synthesis parameters to obtain particular type of properties [16]. CDs can be synthesized via top-down methods which is breaking the bulk materials into nano size, and common examples include chemical oxidation, arc discharge, sonication, electrochemical oxidation, and laser ablation [17–19]. The bottom-up method involves stacking atoms together to form nanosized materials; common examples include thermal carbonization, plasma treatment, microwave synthesis, and hydrothermal synthesis [20–22]. The CDs can be obtained from almost any organic compound, such as banana [23] mango seed [24] wheat flour [25] or any chemical compounds that have C, H, and O bonds present in them, such as ascorbic acid [26] and, citric acid, etc. [27]. A total of 14,716 papers were published on CDs in the last five years, with an increase in the number of papers of up to 149% from 2019 to 2022 according to Scopus statistics up to May 2023.

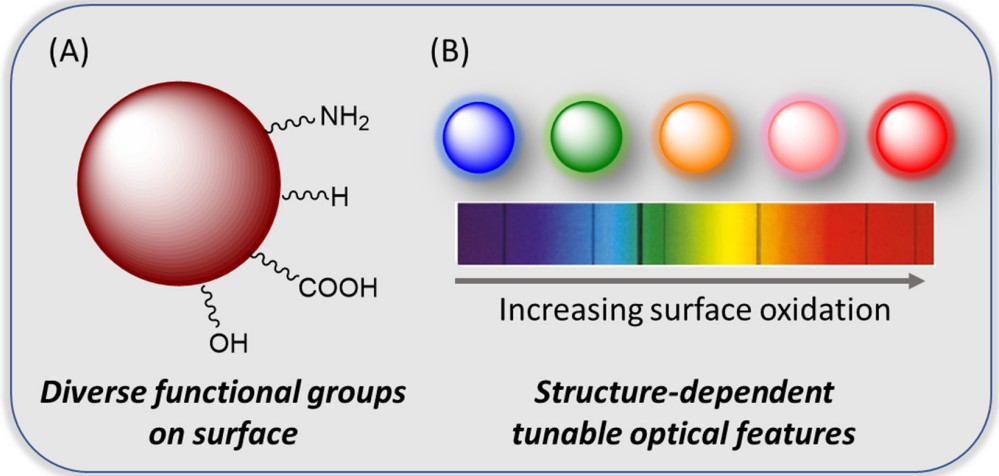

**Figure 1.** Schematic representation of the functional groups that can be formed on carbon nanodots (QDs) (**A**) and their characteristic photoluminescence based on the surface oxidation rates (**B**).

Luminescence is the emission of light at a low temperature. It has different types, such as photoluminescence [28] bioluminescence [29] electroluminescence [30] and cathodo-luminescence [31]. Photoluminescence is caused by the absorption of a photon which causes the excitation of the ground state. If the light is instantaneously emitted by a single excited state (spin orientation does not change), it is called fluorescence. If the delayed emission of light occurs due to the triplet (spin orientation changes) excited state, it is called phosphorescence.

CDs have passivation-based adjustable qualities such as photoluminescence, water solubility, low toxicity, and more. Controlling the properties is, therefore, a critical focus of CD research, and it is possible to accomplish this control provided the factors that affect the properties of CDs are well studied and consistent with others. While some intrinsic factors, such as the type of precursor, determined the final properties of CDs, the role of synthesis parameters, such as time and temperature, is also crucial, as CDs have unique optical properties and are essential for most applications. Therefore, the first step in characterizing CDs is frequently based on optical properties which is to measure the absorption wavelength, that can help to determine the excitation wavelength of CDs. Next, the emission properties can be studied. Figure 2 shows how CDs appear when viewed via a microscope and Figure 2 also demonstrate some important parameters linked to CDs. The Figure 2 demonstrates that the core is primarily made of carbon and it have pores where functional groups are bonded.

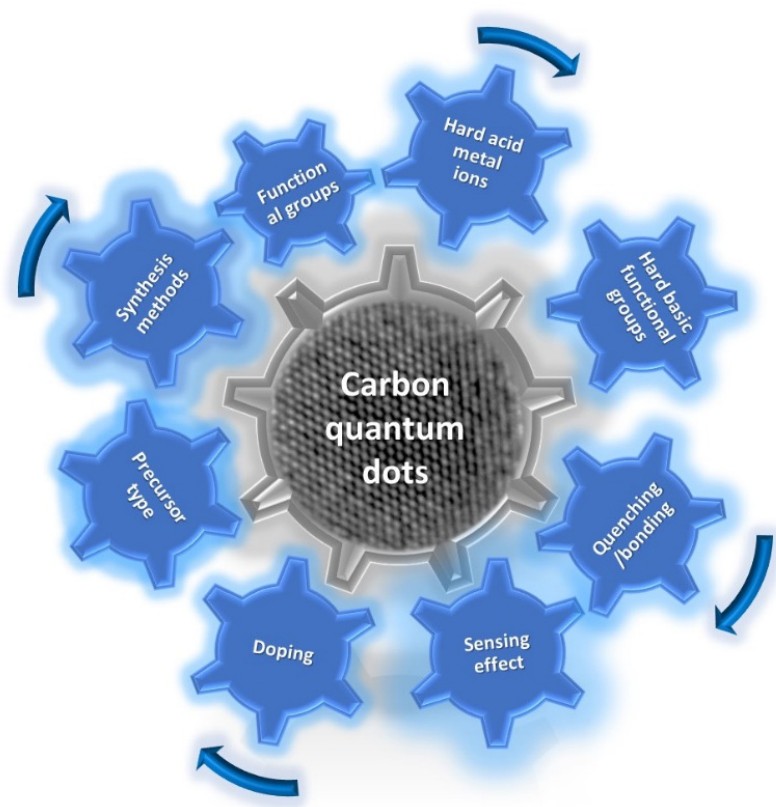

**Figure 2.** Carbon quantum dot mechanism and factors behind them.

Figure 2 clearly illustrates the critical parameters influencing the properties of CDs, such as doping and the sensing effect, and how CD behavior changes in response to these parameters, and highlights all the important features discussed in this review.

## 2. The Metal Ion Sensing Mechanism

The metal ion sensing mechanism is still under debate due to the complex nature of the factors involved in it. One of the proposed mechanisms by researchers is ion bonding. The CDs interaction with metal ions can be detected by means of colorimetric and fluorescence methods. The colorimetric method typically involves using only the human eye. In this method, changes in the color of the solution are observed. Fluorescence spectroscopy is used to observe the emission in the fluorescence method, and changes in the emission spectra observed as a result of some chemical reactions, such as quenching. The change in emission spectra help to observe the behavior of CDs. Quenching or a decrease in the emission spectra of CDs can be caused by energy transfer between CDs and metal ions.

The detection of different metal ions by CDs can be explained via two mechanisms, which are discussed below.

(1) Fluorescence Quenching:

One frequently discussed mechanism used to explain the detection of metal ions with CDs is fluorescence quenching. In this mechanism, the presence of particular metal ions quenches the fluorescence of carbon dots. Several mechanisms, including energy transfer, charge transfer, and the creation of non-fluorescent compounds, can lead to quenching [32]. Metal ions can accept energy or electrons from excited CDs when they come into contact with them, which causes the fluorescence intensity to drop. The concentration of metal ions and their affinity for the CDs surface are two variables that affect how much quenching occurs. Because different metal ions have varying quenching efficiencies, selective detection is possible. [32,33]. Fluorescence quenching can be subdivided into six mechanisms, the details of which will be discussed later in this review paper.

(2)    Metal-Enhanced Fluorescence (MEF):

When the fluorescence of CDs is increased in the presence of particular metal ions, it is called metal-enhanced fluorescence (MEF), and this mechanism is also used to explain the metal ion detection by CDs [34]. The creation of metal-enhanced fluorescence complexes depends on the interaction of the metal ions with the CDs. An enhanced electromagnetic field can be created around the CDs as a result of the metal ions ability to cause localized surface plasmon resonance effects [35]. The CDs radiative decay rate is increased by the strengthened electromagnetic field, resulting in strengthened fluorescence emission. To detect and differentiate particular metal ions, the fluorescence intensification can be tailored. Some other mechanisms such as ligand exchange [36] and redox reaction [37] can also cause the detection of metal ions by influencing the fluorescence spectra.

The fluorescence quenching of CDs with metal ions can be divided into different groups according to various mechanisms, which are described in the following section.

Fluorescence quenching can be caused by static quenching and dynamic quenching [32,38]. Energy transfer can belong of any of the following types: Förster resonance energy transfer (FRET), Dexter energy transfer (DET), surface energy transfer (SET) [38], an inner filter effect (IFE), or photoinduced electron transfer (PET).

Static quenching occurs during the interaction of the quencher with CDs by forming a nonfluorescent complex which, after absorbing light, returns to the ground state. So, it can be said that a complex is formed before the excitation from the ground state takes place [38]. Static quenching decreases with an increase in temperature. In static quenching, no change in fluorescence lifetime occurs, so absorption spectroscopy is frequently used to measure this type of quenching. This phenomenon can detect inorganic materials such as $Hg^{2+}$, $Cu^{2+}$, $Fe^{3+}$, and $Fe^{2+}$, as well as organic materials such as dopamine, nicotinic acid, etc. [39,40].

Dynamic quenching (collisional quenching) is a type of excited-state quenching that occurs when the excited state returns to its ground state after the collision between the CDs and the quencher. There is a change in fluorescence lifetime, so photoluminescence spectroscopy is frequently used to measure this quenching phenomenon. The efficiency of dynamic quenching depends on the concentration of the quencher [32]. The emission spectra of CDs and the absorption spectrum of the quencher do not play any role. Dynamic quenching's presence in the system can be determined by measuring the fluorescence lifetime before and after the reaction [41].

FRET occurs when energy is transferred from the luminescent donor to the energy acceptor within a small distance of 10 to 100 Å. It decreases with an increase in the distance between the acceptor and the donor. It also can be measured by means of fluorescence spectroscopy [42,43].

Dexter energy transfer (DET) is essentiality an electron transfer between a similar redox potential donor and acceptor and occurs if the distance between both is less than 1.5 nm [44], while surface energy transfer (SET) is observed to involve an organic dipole and the metallic surface, and it becomes more prominent if the size of CDs increases from 15 nm [45].

Photoinduced electron transfer (PET) involves CDs and quenchers acting as electron donors or acceptors and forming the cations and anions radical accordingly. In this mechanism, a complex can return to the ground state without photon emission [46,47].

The inner filter effect (IFE) is a nonradiative energy mechanism that occurs due to the absorption of excited–emitted radiation by fluorophores or chromophores in the system [48]. This can occur in two ways:

(1)    One of the quencher absorption spectra overlaps with the excitation spectrum of CDs, so the quencher absorbs light, and quenching take place [49].

(2)    Reabsorption: photons are emitted by one specie and absorbed by others in the solution, and this is due to the weakening of the absorption or excitation radiation by unused quencher and CDs in the solution, so it is not a quenching process by definition [50].

For the CDs with very few functional groups attached to them, the large π-conjugated domains play a central role in the luminescence mechanism [51]. The functional groups on the surface are also called surface defects, which can be trap and emit light of different wavelengths. The number of surface functional groups on CDs depends on the level of oxidation, and high oxidation can cause a redshift of emission wavelength [52,53]. The fluorophores created in the center of the core can also take part in the luminescence mechanism. Quantum confinement arises when it reaches the Bohr radius. Quantum confinement is the change in the conduction band and valence band from a continuous energy level to a discrete energy level. The band gap is inversely proportional to the size of materials when it approaches the Bohr radius. For larger π-conjugated domains, the band gap is small, which causes the redshift in the emission peak. For CDs with long π-conjugated domains and few functional groups, quantum confinement is the major source of luminescence. The π-conjugated domains and photoluminescence properties of CDs can be altered by varying precursor types and synthesis times [54,55].

## 3. Role of Functional Groups in the Metal Ion Sensing Mechanism

Due to their small size, CDs' properties are mostly conceptualized in terms of quantum confinement. However, in reality, almost all of these characteristics depend on the CDs' original precursor and final chemical composition [56,57]. The origin of CDs' photoluminescent behavior is due to their small size, but their chemical characteristics can also influence them. The surface of CDs is covered by a variety of functional groups that are joined to the carbon core structure. Thus, it can be said that CDs' properties depend on their size and functional groups present on the surface [58,59].

The functional groups, which contain $sp^2$ and $sp^3$ hybridized carbons atoms, are located on the surface of CDs. The CDs can also show dichroism (different colors from different angles) due to the complex nature of the functional group. Functional groups originate from oxidation during synthesis and afterward. When the specific absorption wavelength matches with one of the functional group's absorption, it can absorb it, which also leads to multicolor CDs, and so different functional groups can emit light with different wavelengths [60,61]. If the amount of oxygen is increased, the number of functional groups also increases, which traps excitons and causes redshift. Hui Ding et al. [62] reported the emission wavelength from 400 to 625 nm after increasing the oxidation of CDs due to more traps by the functional group. Yuan et al. [63] reported that the color from green to the red of emission changes when the functional group is changed by a solvent. The green emissive CDs have pyridinic N and pyrrolic N groups when measured by FTIR which are not present in red color CDs. Fluorescent molecules which are attached to the core or surface of the CDs can also exhibit fluorescence directly [64].

CDs that can emit light in the whole visible spectrum can open doors for practical applications. There are no such CDs reported until now that can emit full visible spectra. Meng Li Liu et al. [65] reported the synthesis of CDs by means of the Schiff base reaction by utilizing P-benzoquinone and triethylenetetramine (TETA) as a precursor. These CDs can be separated into yellow- and green-color CDs using a silica gel column.

The emergent absorption band of CDs can range from UV to visible range and rarely up to the NIR region [66]. This shift can also be observed during surface functionalization [66] and doping [67]. The absorption involves π–π* aromatic (C=C bonds) transition, which corresponds to a peak from 300 to 400 nm and a peak above 400 nm to n-π* (C=O bond) transition. Arul et al. proposed [68] that the n–π* and π–π* transition is linked with the carbonyl and hydroxyl functional group because the absorption peak changes after adding liquid ammonia to the CDs.

### 3.1. Origin of Optical Properties

Transmission electron microscopy (TEM) reveals the crystallized core structure and amorphous functional group on the surface of CDs. The π-electron system's energy gap transitions possessed intrinsic photoluminescence properties [69]. The energy gap transi-

tions of a $\pi$-electron system exhibit intrinsic PL emission, while the sp$^2$ bonding in CDs also depends on its core size [70]. CDs functional groups can also capture some of the light excitons and contribute to photoluminescence phenomena. It is reported that the oxidation of CDs can enable them to emit different color light in the spectrum from blue to orange [71]. CDs multicolor emissions are associated with their surface defects. A CD's surface has a lot of functional groups, surface defects (broken bonds), and sp$^2$ and sp$^3$ hybridized carbon [72].

Oxygen-abundant functional groups such as hydroxyl, carboxyl, carbonyl, sulfoxide, etc., depend on the oxidation of CDs. These functional groups are attached to CDs core atoms and are the origin of high solubility in polar solvents like water, while also a source of surface defects [73,74]. Rigu Su et al. [75] reported the synthesis of multi-color zinc-doped CDs. They confirmed, using the TEM method, that color change does not depend on CDs size because different emissive colors have a similar size. FTIR and XPS results reveal that zinc doping can decrease the graphic carbon percentage by increasing oxidation, which decreases the other functional group amount. The red emissive CDs have a low number of oxygen-related functional groups, while blue emissive CDs have more oxygen-related functional groups. They use a different ratio of p-phenylenediamine and ZnCl$_2$ precursor to obtain multicolor CDs.

Hydroxyl functional groups (–OH) are mostly present on the surface of CDs due to the organic nature of precursors, and oxidation also contributes to the amount of –OH groups, which was confirmed by FTIR spectroscopy methods. FTIR spectra also confirmed the presence of stretching vibrations and in-plane bending vibrations of -OH [76]. These functional groups have a strong influence on the properties of CDs, while carboxyl groups (–COOH) are also produced when oxidation occurs on a CD's surface. The FTIR spectrum of amino groups (–NH$_2$) produced during the surface functionalization of CDs has been observed, and the these CDs are termed as nitrogen-doped CDs [77].

Nitrogen-based functional groups such as amino, pyridine, nitro, amide, etc., are the second most common functional groups present on the surface and core of CDs. Thanks to the similar atomic radius and electronic structure of carbon and nitrogen, this type of doping is effective [78,79].

Shanshan Wang et al. [80] reported the synthesis of CDs via laser ablation and hydrothermal carbonization methods (HTC). Xylose was used as a precursor and was placed at 200 °C for six hours in an HTC reactor. The CDs obtained from this method are also named HTC to avoid confusion. The HTC CDs were further annealed for two hours at 850 °C with an argon flow rate of 50 cm$^3$/min.

For the laser ablation method, the annealed carbon powder and 20 wt% Teflon powder were used as precursors. The repetition rate was 50 Hz, the pulse width was 1–2 ns, and the produced CDs were named LA-CDs. The hydrothermal carbonization process frequently produces green-emission CDs, while laser ablation produces blue-emission CDs under excitation at 360 nm; although both have similar sizes (4.72 ± 0.7 nm), they show different PL properties. The XPS result shows that oxygen comprising the functional group demonstrates $\pi$–$\pi$* transition, while nitrogen comprising the functional group demonstrates n–$\pi$* and $\pi$–$\pi$* transitions which cause changes in the PL properties of both. They observed that LA-CDs have an N-H bond in their FTIR spectra, while the other two do not have the presence of nitrogen in their FTIR spectra. The LA-CDs had a bright blue emission, whereas the HTC-CDs had a green emission, while annealed CDs did not show any emission when excited by UV. The annealing process probably affects the removal of functional groups. This confirms how vital functional groups are in the phenomenon of CD photoluminescence.

Sulfur-doped CDs are also frequently studied due to similar electron mobility. The sulfur atom has six valence electrons which can alter the optical properties of CDs. The sulfur can attach to the core and the surface of CDs [81], as the functionalization of CDs with thiol can produce an S–H functional group at the peak of 2532 cm$^{-1}$ [82]. The phosphorus-doped CDs are also studied because phosphorus can produce substitutional defects in CDs

surfaces, act as an n-type donor, and alter the optical and electronic properties of CDs [83]. Figure 3 shows the CDs functional group structure comparison of hard acid and soft acid detection using CDs.

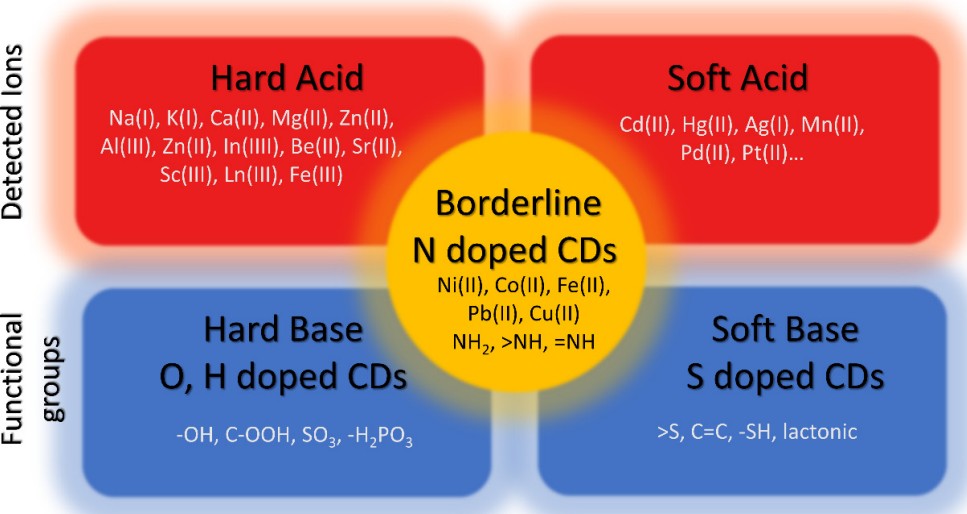

**Figure 3.** Detection of metal ions having a hard or soft character using CDs with diverse surface functional groups.

### 3.2. Mercury Detection

Mercury is one of the heavy metals that can cause significant health problems. It possesses biological toxicity, is non-biodegradable, and has high mobility, which means it stays in the environment for a long time. The Hg–C bond-based organometallic compounds can be present in the air and water for a longer time and cause an imbalance in the biological system, and when ingested by a human, they can potentially damage the nervous system. Because mercury is present in significant quantities near areas where mercury-based minerals are extracted, it is vital to limit the amount of mercury in nature, and thus detecting it in nature is the first step. There are some well-known methods to detect mercury in the samples. Surface-enhanced Raman scattering (SERS), mass spectroscopy, and electrochemical methods are among them. All of these methods require large and complex equipment with an expert operator, limiting their utilization. As already discussed, CDs have good luminescence properties and can form a luminescence turn-on sensor for mercury. The CDs obtained from eggshell membranes are useful, acting as label-free methods to determine mercury, with a detection limit of 2.6 μM [84]. The CDs obtained from L-cysteine were also used as a detecting probe for mercury, and were used with several different metals ions such as $Cu^{2+}$, $Pb^{2+}$, $Ag^+$, $Cd^{2+}$, $Cr^{3+}$, and $Co^{2+}$, but these metal ions do not influence the emission spectra of CDs, even when the concentration is increased up to 10 times, while with mercury, the emission spectra are decreased sharply down to 60% of the total emission spectra [85].

Citric acid monohydrated and ammonia-based nitrogen-doped high luminescent CDs (quantum yield 40.5%) prepared by the one-step hydrothermal method employed as a sensor for $Hg^{2+}$ with a limit of detection up to 0.087 μM, which is 30 times higher than eggshell-membrane-based CDs, and it is suitable in a natural water sample with recovery in the range of 96.6–105.5% [86].

Pineapple-peel-based CDs were prepared by simple hydrothermal treatment and acted as a label-free probe for $Hg^{2+}$ ions. The CDs were utilized to detect $Hg^{2+}$ in lake and tap water samples and found them useful. The logic gate sensor based on NOT operation utilized $Hg^{2+}$ as a input. This shows that CDs can also be used commercially as sensors [87]. A large group of CDs also do not detect mercury ions. For example, hydrothermally



produced CDs using Phyllanthus acidus do not detect $Hg^{2+}$, but can detect $Fe^{3+}$ up to the detection limit of 0.9 μM [88].

### 3.3. Lead Detection

Lead is a toxic transition metal, a non-biodegradable compound, and can react with blood [89]. It can be found in drinking water due to its presence in small amounts in water pipes, and the corrosion of pipes can cause lead to migrate with water. When consumed by humans, even in significantly low amounts of more than 5 mmol/L, it can cause memory loss, mental diseases, and other medical issues [90]. It has three oxidation states, but Pb(II) is mainly found in nature, which can cause mental disabilities, migraine [90], memory loss, and "dullness" [91] in humans, especially children. Different traditional and new techniques are used to detect Pb(II), such as inductively coupled plasma (ICP) mass spectrometry, electrochemical sensors [89], atomic absorption spectroscopy (AAS), DNAzyme [92], and some inorganic nanomaterials. However, all of these methods are expensive, and the demand for new low-cost and sensitive methods is high.

Yeji Kim et al. [82] reported the detection of Pb(II) and explained the quenching mechanism of CDs with Pb(II). They used L-lysine and L-glutathione as precursors to produce CDs via the hydrothermal method. The average lifetime of CDs in PL decay methods decreases to 7.229 ns from 8.753 ns with the addition of Pb(II), which represents about a 19% decrease. They used Pearson's hard and soft acid base principle (HSAB) method to describe the quenching process [82]. Metals ions on the surface of CDs showed that thiophenolate anions could form complexes, and electron transfer can occur from CDs' conduction band to Pb(II). The specific CDs are thermally stable in a natural environment for up to 90 days and have a detection limit of 2.2 μM.

Pooja Chauhan et al. [93] reported the synthesis of CDs from pearl millet seeds by means of the traditional hydrothermal method. These CDs have a high quantum yield of 52% and have a low detection limit of 0.18 nM for $Pb^{2+}$ samples collected from wastewater.

Coccinia Indica-based CDs were reported by K. Radhakrishnan et al. [94] to detect $Pb^{2+}$ with a concentration up to 0.27 μM. The CDs show sensitivity to other metals ions such as $Hg^{2+}$, $Cu^{2+}$, $Pb^{2+}$, and $Fe^{3+}$ ions. When they are passivated with different sensing probes, CDs can act as sensors for different metal ions, so it groups.

CDs derived from sucrose (table sugar) using the microwave heating method for 3 min at 150 °C were used to detect $Pb^{2+}$ by A Ansi et al. [95]. The formation of visible aggregates of CDs in the presence of $Pb^{2+}$ was noted. Negatively charged CDs show the presence of a carboxylate group on the surface and have a good bonding with $Pb^{2+}$ [96]. This was also confirmed by FTIR analysis, in which spectra of C-O related to carboxylate groups were stretched.

Chocolate-derived CDs were able to detect $Pb^{2+}$ at a detection limit up to 12.7 nM [97] via the fluorescence quenching mechanism in the tap water sample. At the same time, tapioca flour-based CDs synthesized via the electrochemical method showed a limit of detection of $Pb^{2+}$ of up to 0.0042 ppm [98].

### 3.4. Silver Detection

Silver is an essential element that has many applications, such as antimicrobial agents in water [99], electrical devices [100], medicine, and electrical devices, and waste related to these applications in the environment is harmful to humans. The recycling of silver is expensive, so it is important to minimize the amount of silver converted into waste [101]. $Ag^+$ ions can change and destroy the healthy nature of pure drinking water, so it is essential to control the number of $Ag^+$ ions in nature. $Ag^+$ ions can be detected using spectroscopic methods such as X-ray fluorescence spectroscopy [102], fluorescence spectrometry, and inductively coupled plasma–atomic emission spectrometry, but these methods require a complex operation. CDs can be used to overcome this situation, which act as probes for $Ag^+$ detection.

Sensing using catalytic properties of CDs produced from a chitosan precursor was reported by Liming Shen et al. [103]. The CDs had a large diameter of $29.4 \pm 6.9$ nm, when mixed with $AgNO_3$ solution the mixture turned yellow, which indicates silver nanoparticle formation. These CDs' ability to act as a catalyst is also confirmed by using chitosan and CDs in dark and light conditions separately with $AgNO_3$ solution, while chitosan shows no activity and the CD solution turned yellow regardless of the presence of light, which shows that CDs act as a catalyst. X-ray photoelectron spectroscopy (XPS) was used to further investigate the phenomenon of silver formation, which shows that the phenol hydroxyl group present on a CD's surface can participate in the growth of nanoparticles. The peak of XPS at 531.7 eV shows -OH of phenol hydroxyl. The FTIR result also shows the presence of phenol hydroxyl groups and amino groups on the CDs' surface, which act as a reducing agent for $Ag^+$ to convert it into elemental silver.

CDs prepared from citric acid and phenazine-2,3-diamine using the solvothermal method were utilized to detect $Ag^+$ in tap water [104]. The detection limit of $Ag^+$ was 31 nM, which is lower than the 0.93 µM value recommended by the world health organization, so this type of CDs can be used to measure the concentration of $Ag^+$. The fluorescence quenching of around 95% occurred within 1 min of mixing, meaning that it can be used as a rapid sensing technique. The mechanism of CDs' fluorescence quenching occurred due to the inner filter effect, electron transfer, static and dynamic quenching, and Förster resonance energy transfer (FRET), but dynamic quenching and FRET decrease fluorescent lifetime, so they are not the leading cause of fluorescence quenching. The absorption spectra confirm the disappearance of the peak at 387 nm when mixing CDs with $Ag^+$, which also excludes the possibility of an inner filter effect and electron transfer. So, static quenching could be the primary reason for this behavior due to nonfluorescent complexes. In static quenching, CDs' fluorescence lifetime does not change when a quencher is added to them, and this quencher's addition also decreases the Stern–Volmer quenching constant with temperature. The zeta potential of $Ag^{2+}$-based CDs is increased to $-3.4$ mV from $-35.3$ mV due to ground state complexes forming when the bonding of -COOH with $Ag^{2+}$ occurs, and the charge obtained by oxygen causes the increase in zeta potential.

Cátia Correia et al. synthesized Eu (III)-doped CDs from citric acid and urea via the hydrothermal method, while they used Eu $(NO_3)_3$ as a europium source [105]. The europium-doped CDs show an increase in the detection performance when compared to undoped CDs due to the difference in the structure of CDs and active sites presence for cations binding. The microscopy images also confirmed that $Eu^{3+}$-doped CDs have a larger size as compared to undoped CDs due to the incorporation of $Eu^{3+}$ ions. Eu (III) can also be joined with the carbon layer due to matching with the carbonyl group of urea and carboxylic groups of the citric acid. Figure 4 shows the quenching effect of $Eu^{3+}$-doped CDs for different metal ions (adapted from open access [105]).

*3.5. Chromium Detection*

Chromium is a highly toxic element found in industrial wastewater. It has two major oxidation states—the non-toxic (at low concentrations) trivalent chromium Cr(III), and hexavalent chromium Cr(VI), which is, even in low amounts, very toxic. Cr(VI) causes cancer and hormonal problems if consumed in sufficient amounts by humans, and its recommended quantity in drinking water is lower than 100 ppb, according to the U.S. Environmental Protection Agency. It can be measured through conventional methods, but fluorescence probes are proven helpful in detecting chromium(VI).

Additionally, CDs exhibit high selectivity and are accessible in use. Researchers are working on the detection of both oxidation states of chromium. MMF Chang et al. reported the synthesis protocol of CDs using a sucrose precursor at a low temperature of 85 °C. These CDs have a yellow color emission. These CDs show pH-dependent fluorescence quenching when treated with Cr(III), which depends on concentration. The limit of detection was found to be $24.58 \pm 0.02$ µM [106].

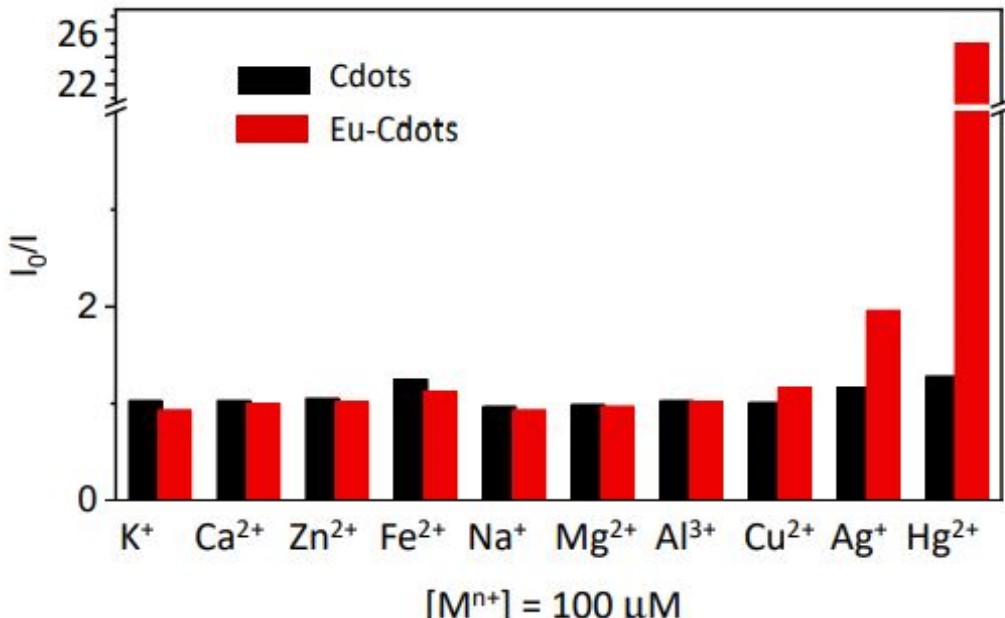

**Figure 4.** Quenching ratio of different ions with $Eu^{3+}$. Quenching by a factor of 2 is observed for $Ag^+$, while quenching by a factor of 25 was observed for $Hg^{2+}$ (adapted from open access [105]).

Qianqian Huang et al. [107] reported blue-emitting CDs synthesized from poria cocos polysaccharide. Those CDs can detect Cr(VI) up to the detection limit of 0.25 mM. They describe the quenching mechanism by comparing and observing the Cr(VI) absorption spectra before and after mixing. The excitation spectra of CDs overlap with the absorption spectra of Cr(VI), due to which the excitation light of CDs is absorbed by Cr(VI), so the inner filter effect occurs, which quenches the fluorescence of CDs. The zeta potential after mixing is also increased from $-18.8$ to $-7.06$ mV due to the binding of the functional group with Cr(VI). In this research, no effect on fluorescence lifetime occurs, so the quenching mechanism was attributed to static quenching.

### 3.6. Iron(III) and Iron(II) Detection

Iron(III) is one the most commonly used metal ions, and is essential for humans up to a certain level; above that level, it can cause diseases such as type 2 diabetes, inflammation [108] and Alzheimer's disease, etc. [109], while a deficiency of iron in the human body can cause anemia (IDA). Iron(III) in the environment also influences plant growth, so it is essential to monitor the quantity of $Fe^{3+}$ in the environment, and more specifically patients with diseases caused by Iron(III) [110].

$Fe^{3+}$ can also cause problems with the production of zinc. During the electrochemical process of zinc production, the efficiency of the process is significantly decreased by iron dissolved in the electrolyte. Ferrite and zinc oxide are involved in the hydrometallurgical production of zinc, and a high temperature is required to remove the ferrite, so it is also essential to control the amount of $Fe^{3+}$ and $Fe^{2+}$ [111].

The detection method for $Fe^{3+}$ is similar to that for other heavy metals, including voltammetry and coupled plasma mass spectrometry (ICP-MS), so now the focus is on detecting iron with the help of a fluorescent probe. Examples of fluorescence probes are conjugated polymer, quantum dot, and CDs, which is also a potential contender for $Fe^{3+}$. $Fe^{2+}$ ions also exist in nature and are essential for humans and must be monitored, but they oxidize to $Fe^{3+}$ in an open environment, making it difficult to detect them accurately. So, there is very little research on $Fe^{2+}$ detection [112].

Hameed Shah et al. [113] reported the synthesis of nitrogen-doped CDs by using nitrogen and carbon-containing precursor N-(2-hydroxyethyl) ethylenediamine tri acetic acid (HEDTA) through the hydrothermal method. Nitrogen-doped CDs have a low quan-

tum yield of 14.17%, and the mechanism is static quenching which occurs in the ground state because PL decreases when the temperature increases. Additionally, PL intensity is independent of $Fe^{3+}$ concentration. This static quenching is caused by -COOH, -OH, -C=O, and -C=N functional groups.

Arpan Bhattacharya et al. [114] reported the synthesis of nitrogen-doped CDs prepared from ethylenediamine (EDA, $\geq$99.5%) which contain about 51.13% carbon, 26.80% oxygen, 5.81% hydrogen, and 16.25% nitrogen. The obtained CDs have a quantum yield of 60.2%. Ferritin is used as a source of $Fe^{3+}$ ions, and electron transfer occurs between CDs and $Fe^{3+}$ vacant d-orbitals, which causes static quenching. They also reported that CDs with a size less than 2 nm could penetrate the ferritin shell and cause more quantum quenching as compared to large-size CDs.

Hao Wu et al. [115] studied the quantum quenching for both nitrogen-doped and boron-doped CDs by using o-phenylenediamine (OPA) and 4-(4,4,5,5-tetramethyl-1,3,2-dioxaborolan-2-yl) benzyl chloroformate as a precursor. $B(OH)_2$ and $NH_2$ functional groups were confirmed by means of FTIR spectroscopy. The limit of detection for (BN-CDs) is 0.1 $\mu$M. In BN-CDs, the fluorescent lifetime remains unchanged after adding $Fe^{3+}$, which indicates static quenching, but does not change for different temperatures, so it can be said that both mechanisms co-occur in coordination with a boron atom with a hydroxyl group to form complexes.

Fanyong Yana [116] reported the inner filter effect in CDs synthesized from anhydrous citric acid (CA) as a source of carbon and diethylenetriamine (DETA) as a source of nitrogen when detecting the $Fe^{3+}$. The limit of detection was found to be 18.11 nM. They used UV-Vis spectroscopy and fluorescence lifetime to determine the inner filter effect.

Pei Zhao et al. used water hyacinth to produce CDs and termed them as wh-CQDs, Produced using the hydrothermal method [117]. They obtained uniform-size CDs and utilized these CDs for the detection of $Fe^{3+}$, and observed a limit of detection of 0.084 $\mu$M, which is even lower than as specified by World Health Organization, which is 0.77 $\mu$M. Quenching of 70% was observed when wh-CQDs were mixed with $Fe^{3+}$, while for other ions, it showed negligible quenching. This phenomenon may be caused by the chelation interaction between the O and N functional groups, and $Fe^{3+}$ ions also have half-filled orbital which can react with excited electrons of CDs, which causes a nonradiative recombination of electrons and holes, resulting in the quenching of photoluminescence. Figure 5 shows the photoluminescence of CDs with different metal ions (adapted from open access [117]).

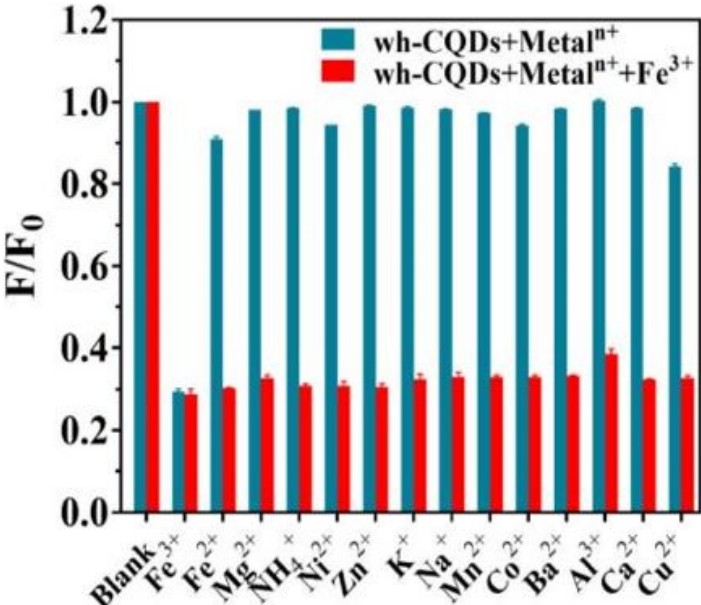

**Figure 5.** The photoluminescence of CDs with different metal ions (adapted from open access [117]).

### 3.7. Copper(II) Detection

Copper ($Cu^{2+}$) is essential for the healthy growth of biological activity because it strengthens [118] the bones and immune system, but an excessive amount of $Cu^{2+}$ can cause vomiting, pain, and disturbance of biological activity [119]. So, it is necessary to develop an easy and inexpensive sensing method to detect $Cu^{2+}$. Researchers are using CDs to detect $Cu^{2+}$ [120–122]. Van Dien Dang et al. [121] synthesized the nitrogen-doped CDs by using citric acid as an oxygen source and ethylenediamine as a nitrogen source. The CDs had a quantum yield of about 84%, and their fluorescence activity decreased after adding different concentrations of $Cu^{2+}$ ions. The limit of detection was observed as 0.076 nM. Xiaochun Zheng et al. [122] described the mechanism of detection of $Cu^{2+}$ based on the functional group by using citric acid and polyethylenimine as precursors for synthesis. The CDs were synthesized by means of the hydrothermal method. The quantum yield was 25% when excited at the wavelength 365 nm. They linked the detection of $Cu^{2+}$ to the presence of amino groups on the surface of CDs, which caused the splitting of $Cu^{2+}$ d-orbital that produced the new path for CDs excited states. The -$NH_x$ group peak CDs in FTIR also disappeared after reaction with $Cu^{2+}$.

CDs can detect metal ions by transferring the electron from the excited state in CDs to metals and then reverting back to the ground state of CDs. As the redox potential plays an important part in the study carried out by Xiaochun Zheng et al., the redox potential play an important factor in metal ion sensing [122]. Redox potential of CDs should also be investigated to further analyze the role of CDs because metal redox potential should be more negative than the holes on CDs and positive than the electrons on CDs. Therefore, measuring the redox potential of different metals ions and comparing it with CDs produced using various precursors can provide further insights.

Xiaoming Li et al. [123] reported the detection of $Be^{2+}$ using CDs. For CD synthesis, citric acid and urea were used as a precursor and heated for different ranges of temperature from 130 to 240 °C. Xiaoming Li et al. mostly focused on CDs created at 160 and 240 °C and named them CDs-160 and CDs-240. From the FTIR spectra, they concluded that CDs-160 have more amino groups due to the visible N–H peak at 1549 $cm^{-1}$. They studied the UV-Vis absorption spectra as shown in Figure 6a,c (adapted from open access [123]) to demonstrate n–π* transition at a wavelength of 234 nm, and a new peak was observed at 335 nm, which causes p–p* transition. This 335 nm peak for 240 degrees has low intensity. CDs-160 exhibited a redshift when compared to CDs-240, while n–π* transition showed a steady peak for both. The CDs-160 also show the high intensity of π–π* transition. They concluded that the amino functional group passivates the trap of CDs-160 with only a single transition mode while CDs-240 are not passivated due to a smaller number of amino groups. They have more surface traps and multiple transition modes.

Figure 6c,d (adapted from open access [123]) show the photoluminescence spectra of CDs-160 and CDs-240. CDs-160 show an excitation-independent peak due to a single transition mode while CDs-240 shows excitation-dependent peaks due to multiple transition modes. So, from here, it can be concluded that CDs with a surface functional group such as carboxyl or hydroxyl groups, etc., show excitation-dependent emission, while CDs that have a high number of amino groups show excitation-independent emission due to the passivation of the surface state by the amino functional group. The functional group has a huge influence on the properties of CDs.

A list of the CD-based metal ion sensing applications reviewed is shown in Table 1, demonstrating the different metal ions discussed in this review paper. Additionally, the detail of the precursors and associated metal ions used during sensing were acquired from the literature are also given in Table 1.

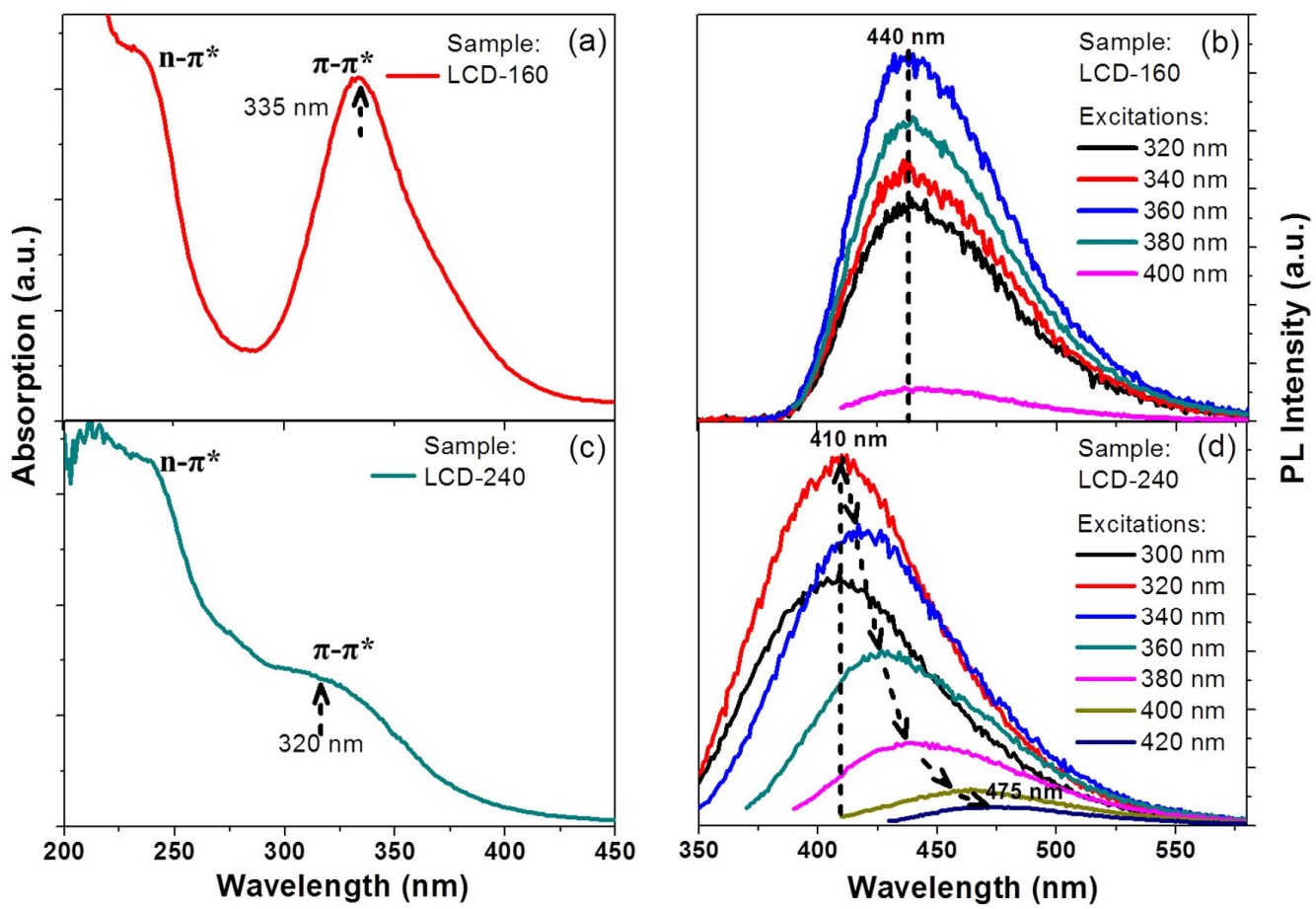

**Figure 6.** (**a**) UV-visible spectra of CDs-160 (**c**) UV-visible spectra of CDs-240 (**b**) PL spectra of CDs-160 (**d**) PL spectra of CDs-240 (Adopted from open access [123]).

**Table 1.** Metal ion detection with carbon dots.

| Precursor | Metal to Detect | Other Metal Ions Used for Detection | Reference |
|---|---|---|---|
| L-lysine and L-glutathione | $Pb^{2+}$ | $Ag^+$, $Ca^{2+}$, $Cd^{2+}$, $Fe^{2+}$, $Hg^{2+}$, $In^{2+}$, $Pb^{2+}$, $Mn^{2+}$, $Ni^{2+}$, $Zn^{2+}$ and $Fe^{3+}$ | [82] |
| Seeds of pearl millet | $Pb^{2+}$ | $Ag^+$, $Cd^{2+}$, $Cu^{2+}$, $Fe^{3+}$, $Co^{2+}$, $Pb^{2+}$, $Zn^{2+}$, $Mg^{2+}$, $Ni^{2+}$, $Ca^{2+}$, $Fe^{2+}$, $Ba^{2+}$, $NH_4^+$, $Na^+$ and $K^+$ | [93] |
| Coccinia indica | $Hg^{2+}$, $Cu^{2+}$, $Pb^{2+}$ and $Fe^{3+}$ | $Ag^+$, $K^+$, $Ca^{2+}$, $Cu^{2+}$, $Ni^{2+}$, $Ba^{2+}$, $Pb^{2+}$, $Hg^{2+}$, $Cd^{2+}$, $Co^{2+}$, $Fe^{2+}$ and $As^{3+}$ | [94] |
| Table sugar | $Pb^{2+}$ | $Cd^{2+}$, $Hg^{2+}$, $Cu^{2+}$, $Fe^{3+}$, $K^+$, $Na^+$, $Ni^{2+}$, $Co^{2+}$, $Cr^{6+}$, $Mn^{2+}$, $Ca^{2+}$ and $Zn^{2+}$ | [95] |
| L-cysteine | $Pb^{2+}$, $Cu^{2+}$ | $Ca^{2+}$, $Fe^{2+}$, $Al^{3+}$, $Pb^{2+}$, $Mg^{2+}$, $Zn^{2+}$, $Fe^{3+}$, $K^+$, $Cu^{2+}$ and $Na^+$ | [124] |
| Chocolate | $Pb^{2+}$ | $Hg^{2+}$, $Fe^{3+}$, $Cu^{2+}$, $As^{3+}$, $As^{5+}$, $Mn^{2+}$, $Zn^{2+}$, $Al^{3+}$, $Mg^{2+}$, $Ni^{2+}$, $Cd^{2+}$, $Co^{2+}$, $Ba^{2+}$, $Ca^{2+}$, $Sn^{2+}$, $Fe^{2+}$, $Ag^+$, $Na^+$ and $K^+$ | [97] |
| Tapioca flour | $Cd^{2+}$, $Pb^{2+}$ and $Cu^{2+}$ | $Mg^{2+}$, $K^+$, $Na^+$, $NO_3^-$ and $SO_4^{2-}$ | [98] |
| Citric acid and phenazine diamine | $Ag^+$ | $K^+$, $Na^+$, $Zn^{2+}$, $Mg^{2+}$, $Ba^{2+}$, $Co^{2+}$, $Ni^{2+}$, $Cu^{2+}$, $Hg^{2+}$, $Pb^{2+}$, $Fe^{3+}$, $Al^{3+}$, $Cr^{3+}$, $As^{3+}$ and $Ag^{2+}$ | [104] |

**Table 1.** *Cont.*

| Precursor | Metal to Detect | Other Metal Ions Used for Detection | Reference |
|---|---|---|---|
| N-(2-hydroxyethyl) ethylenediamine triacetic acid (HEDTA) | $Fe^{3+}$ | $Fe^{3+}$, $Fe^{2+}$, $Ca^{2+}$, $Co^{2+}$, $Cu^{2+}$, $Mg^{2+}$, $Mo^{2+}$, $Zn^{2+}$, $Ni^{2+}$, $Na^+$ and $K^+$ | [113] |
| Sucrose | $Cr^{3+}$ | $Al^{3+}$, $Ca^{2+}$, $Mg^{2+}$, $Co^{2+}$, $Cu^{2+}$, $Cr^{3+}$, $Pb^{2+}$, $Hg^{2+}$, $Ni^{2+}$, $Sn^{2+}$ and $Zn^{2+}$ | [106] |
| Alkali-soluble Poria and Cocos polysaccharide | $Cr^{6+}$ | $Ag^+$, $Ba^{2+}$, $Ca^{2+}$, $Cr^{6+}$, $Cu^{2+}$, $Fe^{2+}$, $Fe^{3+}$, $K^+$, $Mg^{2+}$, $Mn^{2+}$, $Na^+$, $Ni^{2+}$, $Zn^{2+}$ and $Cr^{3+}$ | [107] |
| o-phenylenediamine (OPA) and 4-(4,4,5,5-tetramethyl-1,3,2-dioxaborolan-2-yl) benzylchloroformate | $Fe^{3+}$ | $Ag^+$, $Al^{3+}$, $Ba^{2+}$, $Ca^{2+}$, $Cu^{2+}$, $Fe^{2+}$, $Cd^{2+}$, $Co^{2+}$, $Mg^{2+}$, $Mn^{2+}$, $Na^+$, $Pb^{2+}$, $Sn^{2+}$, $Zn^{2+}$, $Hg^{2+}$ and $K^+$ | [115] |
| Anhydrous citric acid(CA) as a source of carbon while diethylenetriamine (DETA) | $Fe^{3+}$ | $Ag^+$, $K^+$, $Pb^{2+}$, $Cu^{2+}$, $Mn^{2+}$, $Ba^{2+}$, $Ca^{2+}$, $Zn^{2+}$, $Mg^{2+}$, $Hg^{2+}$, $Al^{3+}$, $Fe^{2+}$, $Co^{2+}$, $Ni^{2+}$, $MnO_4^-$ and $Cr_2O_7^{2-}$ | [116] |
| Glucose | $Fe^{3+}$ | $K^+$, $Na^+$, $Ag^+$, $Ca^{2+}$, $Ba^{2+}$, $Cd^{2+}$, $Co^{2+}$, $Cu^{2+}$, $Fe^{2+}$, $Mn^{2+}$, $Ni^{2+}$, $Pb^{2+}$, $Zn^{2+}$, $Hg^{2+}$, $Al^{3+}$ and $Cr^{3+}$ | [125] |
| Ascorbic acid | $Fe^{3+}$ | $Ni^{2+}$, $Co^{2+}$, $Zn^{2+}$, $Mg^{2+}$, $Li^+$, $Fe^{3+}$, $Cu^{2+}$ and $Al^{3+}$ | [126] |
| Urea | $Fe^{3+}$ | C, sugars, $Na^+$, $Mg^{2+}$, $Ca^{2+}$, and $Cl^-$ | [127] |
| Citric acid and ethylenediamine | $Cu^{2+}$ | $Cr^{3+}$, $Mn^{2+}$, $Ni^{2+}$ and $Pb^{2+}$ | [121] |
| Citric acid and Polyethyleneimine | $Cu^{2+}$ | $Na^+$, $Al^{3+}$, $Mg^{2+}$, $Mn^{2+}$, $Li^+$, $K^+$, $Co^{2+}$, $Sb^{3+}$, $Cd^{2+}$, $Zn^{2+}$, $Hg^+$, $Fe^{2+}$, $Fe^{3+}$ and $Cr^{3+}$ | [122] |
| Citric acid and urea | $Be^{2+}$ | $K^+$, $Zn^{2+}$, $Al^{3+}$, $Mn^{2+}$, $Mg^{2+}$, $Cu^{2+}$, $Na^+$ and $Ca^{2+}$ | [123] |

While Table 2 shows that amino groups and nitrogen-based functional groups are able to detect metal ions, irrespective of the nature of metal ions.

**Table 2.** Metal ion detection and functional group of CDs, red color represent hard ion, blue color represent borderline ions while coral color represent soft ions.

| Metal Ion Name and Type | Frequency of Occurrence | Functional Group |
|---|---|---|
| $Be^{2+}$ (Hard) | 1 | Amino-groups |
| $Fe^{3+}$(Hard) | 7 | Nitrogen, Carbon and oxygen based functional group |
| $Cr^{6+}$ (Hard) | 1 | Nitrogen, Carbon and oxygen based functional group |
| $Cr^{3+}$ (Hard) | 1 | Nitrogen, Carbon and oxygen based functional group |
| $Pb^{2+}$ (Borderline) | 7 | Amine, Carboxyl and Thiol, Carboxylate, Hydroxyl and Epoxy |
| $Cu^{2+}$ (Borderline) | 5 | Hydroxy and Amino groups |
| $Ag^+$ (Soft) | 1 | Nitrogen, Carbon and oxygen based functional group |
| $Cd^{2+}$ (Soft) | 1 | Nitrogen, Carbon and oxygen based functional group |

## 4. Summary and Conclusions

In this review, CDs produced using different precursors, synthesis protocols and parameters were compared. It was found that the effect of size is less important than the chemistry of the CD's surface. This leads to several important conclusions:

(1)     Quantum confinement's effect is most prominent if the CDs do not have a large number of functional groups.

(2) Functional groups of hydroxyl and carboxyl are produced via oxidation, while other functional groups such as amine groups are formed due to precursors and solvents from which nitrogen atoms are taken.

(3) The same precursor and synthesis parameters should be used to produce CDs with consistent properties. Based on the review, the obtained CDs' chemical and physical properties are not simply and logically correlated with the synthesis conditions. The relationships are probably not linear. Therefore, slight changes in the synthesis method cause significant differences in the properties, while doping with ions or metals can influence the properties of CDs.

(4) Functional group properties should be investigated separately to determine the effect of CD size on properties. This requires a large amount of work related to the blocking of functional groups, their reduction, or other reactions specific to selected types of functional groups, enabling the understanding of the influence of these groups on the optical and chemical properties of CDs.

(5) Metal ions are detectable with some CDs with which they make complexes and bonds, while not being detectable with other types of CDs. This means that producing different quantum dots specifically sensitive to one metal (or compound) is technically possible. Due to the lack of complete knowledge of CDs' surface chemistry, further research is required.

In preliminary studies with zero-fat milk, we already reported in our research group that amino group-based CDs are dangerous for living organisms, showing antioxidant activity in CDs synthesized from 0% milk fat, which is controlled by their concentration [128]. As the functional group plays an important role in metal ion detection, so other applications involving chemical reactions such photodegradation a bio-sensor functional group should be further investigated.

More thorough research is required to comprehend the chemical characteristics of CDs. It is important to highlight the fact that CDs exhibit very high quantum yields. They have a quantum yield efficiency comparable to or higher than quantum dots based on Cd, Se, Te, etc. There are many novel applications that require the production of low-toxicity devices, e.g., QLED TVs.

**Author Contributions:** Conceptualization, H.S.; formal analysis, H.S. and M.W.; investigation; resources, M.W.; data curation, H.S.; writing—original draft preparation, H.S.; writing—review and editing, H.S., E.C. and M.W.; supervision, M.W.; project administration, M.W.; funding acquisition, M.W. All authors have read and agreed to the published version of the manuscript.

**Funding:** Research financed under the "university grant" qualified for funding in the 4th (2023) edition of the competition "System of university grants for research carried out with the participation of doctoral students" (Action 4 in the Project "Initiative of Excellence—Research University" at AGH) entitled "Synthesis and functionalization of carbon quantum dots obtained by hydrothermal method" Application No. 6438.

**Data Availability Statement:** Not applicable.

**Conflicts of Interest:** The authors declare no conflict of interest.

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
