# Peer review of "Carbon Quantum Dots: The Role of Surface Functional Groups and Proposed Mechanisms for Metal Ion Sensing"

_inorganics, doi:10.3390/inorganics11060262_

Round 1

Reviewer 1 Report

1.          Language check is highly recommended as there are too many problems in grammar. Just a few examples:

The Figure 1 (B) shows oxidation causes ….” The is not necessary.

“Although C, H, and O are the main component of CDs, however, ? also can have some other elements like Nitrogen ….” Missing subject.

“The colorimetric method is frequently can be a naked eye method …” Redundant verbs.

“…..the increase or decrease in emission spectra is also rarely observed.”  Using a negative adverb for a positive expression. Suggest change to “sometimes”.

“….and it becomes more prominent if the size of CDs increases from 15nm”  suggest change to is greater than.

There are many more that need correction.  Also pay attention to typographical error like “… organic nature of precursors and oxidation also contribute in the amount of –OH groups which was confirmed by FTIR spectroscopy methods_,spectra also …”.

2.          Some of the terms used to characterize bonding nature need more clear definition.  Hard and soft acid base, for example should have a tangible measure¸ especially on the metal ion side. 

3.          As a review article, this manuscript can be more critical.  For example, many different CDs are cited but only a small common group of metal ions are detectable with these CDs. On one hand, the fact that the majority of metal ions are not detectable is a limitation of CDs as metal sensor. On the other hand, there could be a reason why the majority of CDs shows a common affinity for this small group of metals. The authors actually have more important subjects to write about.

4.          If Figure 4 has an official permission for adoption, the authors should state so.

5.          CDs is actually an organic material, especially when people are no longer doing a lot of top-down synthesis. This article would be more appropriate if more information is added to suit the scope of Inorganics in synthesis, characterization, and spectroscopies.

Author Response

Dear Reviewer,

I hope this letter finds you well. I appreciate your effort in reviewing our review paper on the topic of “Carbon quantum dots: the role of surface functional groups and proposed mechanisms for metal ion sensing”.I'm happy to let you know that we carefully considered your recommendations and made the required edits to the manuscript. Your constructive feedback has helped us improve our method, the substance, and the clarity and coherence of our conclusions. I think these changes significantly improved this research's overall accuracy and significance. All the answers are given below

Best Regards

Reviewer 2 Report

The manuscript by Shabbir et al. reviewed the role of the functional groups in the properties of carbon dots. The report also provides some insights into metal ion sensing. The topic is interesting and meaningful. One of the main drawbacks to this manuscript would be the disorganized structure, which requires extensive modification to make it attractive to people who would like like to learn about this research field. Nonetheless, I believe that it might be suitable for publication in Inorganic after major revisions.

1. The Abstract is a little bit confusing and needed to be rewritten. It should provide a condensed summary and significance of this review.

2. For each section, the paragraph is very rich without figures. For instance, 

Page 3, the author was trying to introduce many types of ion sensing mechanisms and concepts, it is so dry to read this large body of TEXT, it would be better to use some schematic representations. Same issue is found on page 6.

3. What is the author trying to convey in Figure 2?

4. In Sections 3.2 to 3.7, the author categorizes the references by different ions, however, they narrate in a simple direct way without any discussions. The author should connect the specific references with the concept and ion-sensing mechanisms introduced in the previous paragraphs.

Extensive editing of the English language is required.

Author Response

Dear Reviewer,

I hope this letter finds you well. I appreciate your effort in reviewing our review paper on the topic of “Carbon quantum dots: the role of surface functional groups and proposed mechanisms for metal ion sensing”.I'm happy to let you know that we carefully considered your recommendations and made the required edits to the manuscript. Your constructive feedback has helped us improve our method, the substance, and the clarity and coherence of our conclusions. I think these changes significantly improved this research's overall accuracy and significance. All the answers are provided in the file 

Best Regards

Reviewer 3 Report

The manuscript provides an extended overview of the current state and progress in the synthesis of carbon dots and their applications for the detection of metal ions. The review is clearly structured and well readable. It includes definitions of the optical properties on which the authors focus their analysis in the review.

My main remarks relate to formatting and presentation, but not solely.:

1. The font of the text looks different for the main body and the references.

2. Page 10, chapter 3.7, paragraph 2 and following - formatting of spaces is required (there are missing and extra ones).

3. Tables 1 and 2 need alignment and formatting. The contents of the tables are not sufficiently described in the text. The entire chapter 3.7 requires a thorough formatting check.

4. Punctuation throughout the manuscript needs checking.

5. The names of journals are missing in most of the references.

6. References in the main text are presented in two different ways - superscript and in square brackets - they should be made consistent.

7. Punctuation must be checked.

8. Confusing and long sentence on page 13, last paragraph:

"As functional group plays important role in metal ion detection so for other application involves chemical reaction like photodegradation and as a bio-sensor functional group should be further investigate and in the preliminary studies with zero fat milk we already reported in our research group that amino group based CDs are dangerous for living organism which show antioxidant activity in carbon CDs synthesized from 0% milk fat, that is controlled by their concentration and synthesis time.. "

9. Page 11, the phrase “The CDs based metal ions sensing reviewed are showed in Figure 4.” Is this really what the authors wanted to say? If so, it is not clear from Figure 4.

Author Response

Dear Reviewer,

I hope this letter finds you well. I appreciate your effort in reviewing our review paper on the topic of “Carbon quantum dots: the role of surface functional groups and proposed mechanisms for metal ion sensing”.I'm happy to let you know that we carefully considered your recommendations and made the required edits to the manuscript. Your constructive feedback has helped us improve our method, the substance, and the clarity and coherence of our conclusions. I think these changes significantly improved this research's overall accuracy and significance. All the answers are provided 

Best Regards

Round 2

Reviewer 1 Report

none

Reviewer 2 Report

I do not have any further concerns and recommend acceptance.

No further comments